# Sudden Onset of IgA Vasculitis Affecting Vital Organs in Adult Patients following SARS-CoV-2 Vaccines

**DOI:** 10.3390/vaccines10060923

**Published:** 2022-06-09

**Authors:** Yunjung Choi, Chang Hun Lee, Kyoung Min Kim, Wan-Hee Yoo

**Affiliations:** 1Division of Rheumatology, Department of Internal Medicine, Jeonbuk National University Medical School, Jeonju City 54907, Korea; imcyj@jbnu.ac.kr; 2Research Institute of Clinical Medicine of Jeonbuk National University-Biomedical Research Institute of Jeonbuk National University Hospital, Jeonju City 54907, Korea; chleegi@jbnu.ac.kr (C.H.L.); kmkim@jbnu.ac.kr (K.M.K.); 3Division of Gastroenterology, Department of Internal Medicine, Jeonbuk National University Medical School, Jeonju City 54907, Korea; 4Department of Pathology, Jeonbuk National University Medical School, Jeonju City 54907, Korea

**Keywords:** IgA vasculitis, COVID-19, vaccination, ChAdOx1, mRNA-1273

## Abstract

IgA vasculitis is an immune complex-mediated small-vessel vasculitis that mainly occurs in children and is characterized by palpable purpura, arthralgia, abdominal pain, and glomerulonephritis. We report three cases of new-onset IgA vasculitis involving major organs in adult patients after they received either the ChAdOx1 viral vector (Oxford/AstraZeneca) vaccine or the messenger RNA-1273 (Moderna) vaccine. These cases suggest that COVID-19 vaccines have the potential to trigger IgA vasculitis and indicate that physicians need to monitor for this possible complication.

## 1. Introduction

COVID-19 vaccination has been recommended worldwide as one of the most effective ways to reduce the complications of the SARS-CoV-2 virus. However, few studies have been conducted regarding the side effects of vaccination, and thus vaccination programs have been implemented without established adverse events profiles. Due to the heterogeneous pattern of vaccine side effects and difficulty establishing the correlation between COVID-19 vaccination and autoimmune adverse effects, these phenomena have been overlooked or remained undetected.

Recently, there have been increasing reports of immune-mediated inflammatory diseases after individuals received a COVID-19 vaccine. The post-vaccinal diseases and autoimmune disorders reported include systemic lupus erythematosus, systemic sclerosis, anti-neutrophil cytoplasmic antibody-associated vasculitis, inflammatory myopathy, and autoimmune hemolytic anemia.

There are two types of COVID-19 vaccines authorized for use to prevent the spread of COVID-19: messenger (m) RNA-based vaccines (BNT162b2; Pfizer/BioNTech, mRNA-1273; Moderna) and viral vector vaccines (ChAdOx1; Oxford/AstraZeneca, Ad26.COV2.S; Janssen).

Among the autoimmune disorders that have been reported to follow vaccination, IgA vasculitis has attracted research attention as the number of reported cases rises.

Herein, we report three cases of new-onset IgA vasculitis in adults that received either a viral vector vaccine or mRNA-based vaccine. Unlike the excellent prognosis in most IgA vasculitis cases, the patients we observed presented with aggressive courses that affected several major organs.

## 2. Case 1

Our first patient, a 60-year-old male, developed a purpuric rash on his upper and lower extremities, abdomen, and lower back, 10 days after receiving the second dose of the ChAdOx1 viral vector (Oxford/AstraZeneca) vaccine. He had no history of underlying illness. He presented at our emergency room several days afterwards with abdominal pain and melena. At the time of his visit, enteritis and mesenteric panniculitis were detected by his abdominal CT scan. His urinalysis showed hematuria (RBC count 21–50/HPF) and a urine protein–creatinine ratio (UPCR) of 3902 mg/g. He was prescribed prednisolone of 1 mg/kg/day and his symptoms improved with this course of treatment. His abdominal pain and skin eruption resolved, and 1 week after the therapy began, his level of C-reactive protein (CRP) had decreased from 131.5 mg/dL to 8.14 mg/dL. After 3 months, the prednisolone was tapered to 2.5 mg/day, and the patient’s proteinuria and hematuria went into remission. The patient continues to attend regular follow-up appointments and the steroid prescription will be discontinued in the near future.

## 3. Case 2

Our second patient, a 63-year-old male, complained of abdominal pain, purpura on both lower legs, and arthralgia, all of which developed 1 day after receiving the first dose of the mRNA-1273 (Moderna) vaccine. He reported he did not have any medical history of chronic diseases. He showed gross hematuria (RBC count 11–20/HPF), proteinuria (UPCR 512 mg/g), and a decreased glomerular filtration rate of 64 mL/min/1.7 m^2^. An abdominal CT scan, which was performed due to abdominal colicky pain, revealed wall thickening of the terminal ileum, which suggested enteritis. As soon as a diagnosis of IgA vasculitis was made, the patient was administrated with prednisolone of 1 mg/kg/day. With the treatment, his arthralgia, abdominal pain, hematuria, and proteinuria showed clinical improvement. His CRP level decreased from 66.6 mg/L at the time of admission to 5.95 mg/L at 1 week after admission. As the patient’s general condition improved, he wanted to be discharged from the hospital as soon as possible. Due to his demand, he was discharged with a prescription, despite the concerns of the medical staff of insufficient time in medical care and the increased risk of relapse. Two weeks after discharge, he visited the emergency room with complaints of melena. A hemoglobin level of 7.1 g/dL was detected and a follow-up abdominal CT scan showed markedly increased small bowel wall thickening compared to the initial CT scan performed 2 weeks prior. An endoscopy and colonoscopy were also performed, which did not identify any active bleeding sites in the stomach or colon. Based on a diagnosis of small bowel bleeding due to a relapse of IgA vasculitis, he was re-admitted to our clinic, administered steroids intravenously, and began a therapeutic fasting regimen. Following the treatment and rest, no signs of bleeding were detected. As clinical progress had improved, he was discharged after sufficient treatment with a tapered dose of prednisolone. The patient continues to take low-dose steroids on a regular basis.

## 4. Case 3

Our third patient, a 74-year-old male, was referred to our emergency room 14 days after receiving the second dose of the ChAdOx1 viral vector (Oxford/AstraZeneca) vaccine, with a purpuric rash on his lower extremities, as well as myalgia, arthralgia, and melena. He did not have any medical history of chronic illness except benign prostatic hyperplasia. The secondary hospital where he was first admitted administered 125 mg/day of methylprednisolone intravenously for 3 days and referred the patient to our tertiary hospital due to his aggravated symptoms. When he visited our emergency room, he presented with overt hematuria (RBC count over 50/HPF) and proteinuria (UPCR 4771 mg/g). His creatinine level was 1.21 mg/dL, WBC count was 15,830/uL, and CRP was 55 mg/L. The spread of palpable purpura on the extensor arm, legs, and abdomen was also detected. His abdominal CT scan showed acute enteritis from the jejunum to the proximal ileum with a small amount of ascites. Based on clinical symptoms and indicators, he was diagnosed with IgA vasculitis. He was prescribed 125 mg/day of intravenous methylprednisolone for 5 days and 1 mg/kg/day of prednisolone after the methylprednisolone treatment ended. After 2 weeks of treatment, his skin rash, myalgia, and arthritis showed improvement; however, his reduced level of renal function was not restored to normal. A renal biopsy was performed to exclude other renal diseases, and the histopathology, which showed immunofluorescent IgA deposits in the mesangium, was consistent with IgA nephropathy (Figure 1A,B). He was discharged in a stable condition with a normalized CRP level and improved clinical symptoms, with the exception of decreased renal function (GFR of 48 mL/min/1.7 m^2^). He was prescribed a tapered dose of prednisolone (0.5 mg/kg/day) and azathioprine and attended follow-up appointments with the outpatient department (OPD) every other week for 1 month. However, 1 month after discharge, he complained of abdominal pain and pretibial pitting edema. He was admitted for edema treatment and was prescribed diuretics. A few days after admission, he presented with massive hematochezia and altered vital signs. The patient’s hemoglobin count was 4.7 g/dL, systolic blood pressure was 70 mmHg, and pulse rate was 100 bpm. To promote hemostasis, a transarterial embolization was attempted, but the procedure failed. Thus, an exploratory laparotomy was performed. We observed a mesenteric hematoma of the proximal ileum, and both wall thickening and nodularity were noted in the lumen of the ileum. A segmental resection of the proximal to mid ileum was performed. A histopathological analysis of the ileum showed diffuse neutrophil infiltration in the submucosal layer, hemorrhaging, necrosis, and extensive fibrin deposition, which corresponded with the diagnosis of IgA vasculitis (Figure 1C,D). The up-titration of prednisolone dose to 1 mg/kg/day was prescribed to him. At first, a prescription of the immunosuppressant cyclophosphamide was considered; however, considering its potential side effects and the patient’s age, mycophenolate mofetil was prescribed instead. After a small bowel resection and treatment with a high-dose prednisolone and mycophenolate mofetil, he no longer presented any signs of bleeding. Two weeks after the small bowel resection, the patient was discharged without further complications. He was prescribed mycophenolate mofetil and low-dose prednisolone and attended follow-up appointments every 3 weeks for 3 months.

## 5. Discussion

IgA vasculitis is an immune complex-mediated small-vessel vasculitis characterized by IgA1-dominant immune deposits in vessel walls [1]. Typically, IgA vasculitis occurs after an upper respiratory tract infection, although the pathogenesis of IgA vasculitis is still not completely understood. The symptoms of IgA vasculitis include palpable purpura, arthralgia, abdominal pain, and glomerulonephritis [2]. This illness usually presents in children, but it also affects adults with an incidence of 0.8–2.2 per 100,000 person-year [3]. Furthermore, it was previously reported that older patients showed an increased risk of a severe course of disease, including glomerulonephritis, compared to younger patients. Most cases of IgA vasculitis have a favorable prognosis, but 10–40% of IgA vasculitis cases involve the gastrointestinal system and 10–55% involve the kidneys [4,5]. Renal complications are known as the main cause of morbidity and mortality in individuals suffering from IgA vasculitis [6].

In this report, we introduced three cases of new-onset IgA vasculitis following COVID-19 vaccination (Table 1). These patients presented with major organ involvement such as the gastrointestinal system and the kidneys, as well as constitutional symptoms including mucocutaneous manifestations and arthralgia. Two cases developed after receiving a viral vector vaccine (Oxford/AstraZeneca, ChAdOx1) and one case developed after the mRNA-based vaccine (Moderna, mRNA-1273). The relationship between these occurrences of IgA vasculitis following vaccination against COVID-19 dominantly favors correlation rather than coincidence, based on other cases which have been recently reported.

There have been increasing reports of cases of IgA vasculitis that occurred after COVID-19 vaccination [7,8,9,10,11,12]. Table 2 summarizes the cases of new-onset or relapsed IgA vasculitis following COVID-19 vaccination by conducting a searching of the Pubmed as of 31 May 2022. Wu et al. analyzed 12 cases of IgA vasculitis that occurred after COVID-19 vaccination; six cases were new onset and the other six were relapses [8]. The patients received an mRNA-based vaccine (Pfizer-BioNTech or Moderna vaccine), and macrohematuria was observed in two-thirds of the cases. Crescentic IgA nephropathy developed in two patients, and both were treated with a high dose of steroids and cyclophosphamide. With supportive care, the remaining 10 patients’ hematuria eventually resolved. The occurrence of IgA vasculitis has also been reported following vaccination with a viral vector vaccine, such as ChAdOx1 of Oxford/AstraZeneca. The report by Sugita et al. presented three cases of new-onset IgA vasculitis, in a 72-year-old male, 62-year-old-male, and 76-year-old-female, after they received a viral vector vaccine (ChAdOx1, Oxford/AstraZeneca). They showed hematuria but did not present with proteinuria or significantly decreased renal function. The 20–40 mg/day of prednisolone or methylprednisolone was prescribed to the patients; two of them showed complete remission, and the data concerning the prognosis of the third patient was not available [12]. These documented cases suggest that the development of IgA vasculitis after COVID-19 vaccination is not limited to coincidence, and that the vaccines have the potential to trigger IgA vasculitis.

The pathogenesis of IgA vasculitis after COVID-19 vaccination is not fully understood, but some pathogenic mechanisms have been hypothesized. A possible explanation is that the increased anti-glycan antibodies elicited by the COVID-19 vaccines cross-react with galactose-deficient IgA1, which are the essential molecule in the multi-hit mechanism of the pathogenesis of IgA vasculitis, and they form immune complexes and deposits in small vessels [8,13,14]. In previous studies, IgA vasculitis with nephritis showed higher serum levels of galactose-deficient IgA1-specific antibodies compared to healthy control subjects and IgA vasculitis cases without renal involvement [15]. Based on these previous findings, we speculate that our patients who showed significant renal and bowel involvement had high levels of galactose-deficient IgA1-specific antibodies. In addition to the antigen-antibody complex-mediated process, other hypotheses for the possible immunopathology of COVID-19 vaccine-induced IgA vasculitis include inappropriately potentiated Th2 responses, and hyperactivation of autoreactive B cells [7,8]. Interestingly, the development of IgA vasculitis in SARS-CoV-2-infected patients has recently been reported [16,17,18]. These findings strongly imply that COVID-19 vaccinations could trigger the development or recurrence of IgA vasculitis.

Although these newly reported cases implicate COVID-19 vaccines in the development of IgA vasculitis, there has not been an authoritative large-scale study to verify this association. Due to the difficulty in proving the correlation between the vaccines and IgA vasculitis, the possibility of developing IgA vasculitis after COVID-19 vaccination may be underestimated. However, considering recent reports and the hypothesis, immunopathology of COVID-19 vaccine-induced IgA vasculitis and the mass vaccination campaign against COVID-19, in particular, plans to vaccinate school children and infants, the number of IgA vasculitis cases is expected to increase. Thus, it is essential to add the COVID-19 vaccines to the list of triggers of IgA vasculitis. Furthermore, guidelines should be established to respond to instances in which patients have experienced vaccine-induced IgA vasculitis, regarding whether they should receive a vaccine booster shot. In addition, further research is needed including the clinical differences between regular IgA vasculitis cases and onset after vaccination, the outcome of COVID-19 vaccination-related IgA vasculitis, as well as the baseline and serial serum level of galactose-deficient IgA1.

**Table 2 vaccines-10-00923-t002:** A summary of new-onset or relapsed IgA vasculitis following COVID-19 vaccination.

Author, the Year of Report	Age/Gender	Underlying Disease	Type of Vaccination	New Onset or Relapse	Time to Present Symptom after Vaccination (Days)	Vaccine Dose	Proteinuria (UPCR, g/g)	Hematuria	Other Clinical Symptoms	Treatment	Outcome
Anderegg et al., 2021 [19]	39/M	HTN	mRNA-1273 (Moderna)	N	0	2	NA	Macrohematuria	Fever, flu-like symptom	PD, CYC	Hematuria persisted
Abramson et al., 2021 [20]	30/M	None	mRNA-1273 (Moderna)	N	1	2	0.8	>30 cells/HPF	Fever, headache	RAASi	Proteinuria reduction
Kudose et al., 2021 [21]	50/F	HTN, Obesity, anti-phospholipid syndrome	mRNA-1273 (Moderna)	N	2	2	2.0	>50 cells/HPF	Generalized weakness, decreased appetite	NA	Hematuria resolved
Kudose et al., 2021 [21]	19/M	Microhematuria	mRNA-1273 (Moderna)	N	2	2	None	Numerous red blood cells	NA	NA	Hematuria resolved
Park et al., 2021 [22]	22/F	IgA vasculitis	mRNA-1273 (Moderna)	R	2	2	0.4	>50 cells/HPF	None	Supportive	Hematuria resolved
Park et al., 2021 [22]	39/F	None	mRNA-1273 (Moderna)	NA	2	2	0.9	>50 cells/HPF	None	Supportive	Hematuria resolved
Park et al., 2021 [22]	50/M	Chronic kidney disease	mRNA-1273 (Moderna)	NA	1	2	3.56	>50 cells/HPF	None	RAASi	Hematuria resolved
Park et al., 2021 [22]	67/M	Chronic kidney disease	mRNA-1273 (Moderna)	NA	1 month	1	2.90	>50 cells/HPF	Purpuric rash	PD	Hematuria resolved
Perrin et al., 2021 [23]	22/M	IgA nephropathy	mRNA-1273 (Moderna)	R	2 and 25 after first dose, 2 after second dose	2	0.34	Gross hematuria	Arthralgia	NA	Hematuria resolved
Grossman et al., 2021 [24]	94/M	None	mRNA-1273 (Moderna)	N	10	2	3+	3+	Purpuric rash	PD	Hematuria resolved
Negrea et al., 2021 [25]	38/F	IgA nephropathy	mRNA-1273 (Moderna)	R	Several hours after	2	1.4	Micro hematuria	Body ache, fatigue, headache	NA	NA
Negrea et al., 2021 [25]	38/F	IgA nephropathy	mRNA-1273 (Moderna)	R	Several hours after	2	0.4	Micro hematuria	Body ache, fatigue, headache	NA	NA
Obeid et al., 2021 [7]	78/F	IgA vasculitis	mRNA-1273 (Moderna)	R	7	1	NA	150 × 10⁶/L	Diarrhea, Abdominal pain	MP	Improved rapidly
Perrin et al., 2021 [23]	27/F	IgA nephropathy, Hemodialysis	BNT162b2 (Pfizer-BioNTech)	R	2	2	1.9	Gross hematuria	Abdominal pain, Urticaria	NA	Hematuria resolved
Perrin et al., 2021 [23]	41/F	IgA nephropathy, kidney transplantation	BNT162b2 (Pfizer-BioNTech)	R	2	1	0.47	Gross hematuria	None	NA	Hematuria resolved
Maye et al., 2021 [26]	23/M	NA	BNT162b2 (Pfizer-BioNTech)	R	1	2	UACR 4.9 mg/mmol	165 cells/mm^3^	Purpuric rash	PD	Hematuria resolved
Mohamed et al., 2021 [27]	50/M	Seasonal allergy	BNT162b2 (Pfizer-BioNTech)	N	14	1	1.1	10 cells/HPF	Purpuric rash	PD, RAASi	Decreased UPCR of 0.5 g/day
Sirufo et al., 2021 [11]	76/F	None	ChAdOx1 (Oxford/AstraZeneca)	N	7	1	Neg	72 cells/HPF	Purpuric rash	Deflazacort	Hematuria resolved

CYC, cyclophosphamide; HPF, high-power field; MP, methylprednisolone; N, new; NA, not available; PD, prednisone; R, relapse; RAASi, renin-angiotensin-aldosterone system inhibitors; UACR, urine albumin–creatinine ratio; UPCR, urine protein–creatinine ratio.

## 6. Conclusions

Even though the COVID-19 vaccine is currently the most effective preventative measure against the SARS-CoV-2 virus, delay and refusal to receive the vaccine due to the occurrence of adverse reactions has become an issue [28]. Most cases of IgA vasculitis have a self-limited natural course, but refractory and severe cases, as seen in our study, can also occur. Monitoring the safety of the COVID-19 vaccines, and providing the correct information about their benefits and potential adverse reactions, are essential to gaining the public’s trust. Thus, it is critical for physicians to monitor for COVID-19 vaccination-related IgA vasculitis during post-vaccine health care management.

## Figures and Tables

**Figure 1 vaccines-10-00923-f001:**
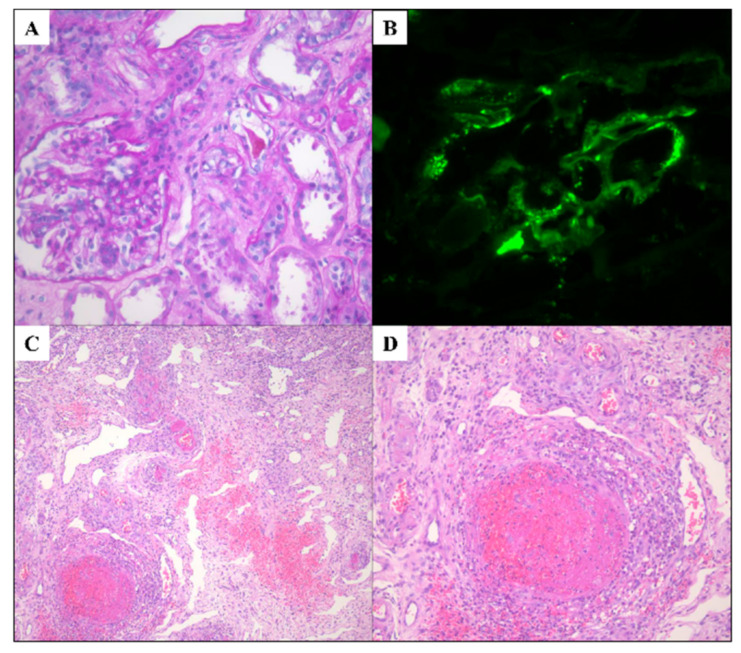
Histologic features of kidney and ileum in a patient with IgA vasculitis after COVID-19 vaccination. (**A**) Glomerulus showing mesangial hypercellularity and mesangial matrix expansion (PAS stain, original magnification: ×400). (**B**) Immunofluorescent staining reveals IgA deposits in mesangium (original magnification: ×400). (**C**) Diffuse inflammatory infiltration in submucosal layer of ileum, and fibrinoid necrotizing vasculitis in several vessels (H&E stain, original magnification: ×100). (**D**) Fibrinoid necrotizing vasculitis of ileum (H&E stain, original magnification: ×400).

**Table 1 vaccines-10-00923-t001:** Characteristics of the patients with IgA vasculitis after COVID-19 vaccination.

Patient Characteristics	Patient 1	Patient 2	Patent 3
Age	60	63	70
Sex	M	M	M
Type of vaccination	ChAdOx1 (Oxford/AstraZeneca)	mRNA-1273 (Moderna)	ChAdOx1 (Oxford/AstraZeneca)
Time to present symptoms	10 days	1 days	14 days
Baseline Cr level (mg/dL)	0.76	0.67	0.59
Peak Cr level after COVID-19 vaccine (mg/dL)	0.98	1.29	3.34
Gross hematuria before COVID-19 vaccine	0–2/HPF	0–2/HPF	0–2/HPF
Gross hematuria after COVID-19 vaccine	21–50/HPF	21–50/HPF	21–50/HPF
Proteinuria before COVID-19 vaccine	Neg	+	+
Peak UPCR after COVID-19 vaccine (mg/g)	3902	2214	15764
UPCR 2 months after COVID-19 vaccine (mg/g)	584	454	6618
Other clinical symptoms	Purpura, Abdominal pain, Melena	Purpura, Abdominal pain, Melena, Arthralgia	Purpura, Abdominal pain, Melena, Arthralgia
Current treatment	PD	PD	MMF, PD

Cr, creatinine; HPF, high-power field; MMF, mycophenolate mofetil; PD, prednisone; UPCR, urine protein/creatinine ratio.

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
