# Peer review of "Sudden Onset of IgA Vasculitis Affecting Vital Organs in Adult Patients following SARS-CoV-2 Vaccines"

_vaccines, 2022, doi:10.3390/vaccines10060923_

Round 1

Reviewer 1 Report

In this interesting manuscript, Yunjung Choi and his colleagues reported three cases of new-onset IgA vasculitis that affected vital organs after they received either the ChAdOx1 viral vector or mRNA-based SARS-CoV-2 vaccination. 

The authors described the patients'  clinical characteristics, medical history, treatment, and clinical response. There have been several reports of IgA vasculitis following COVID-19 vaccination (vector-based and mRNA vaccines). However, the relevance between IgA vasculitis and vector-based mRNA-based vaccination needs more cases and additional research. These case reports add additional cases to this pool of evidence. Although it is still unclear whether the SARS-CoV-2 vaccine triggers IgA vasculitis, earlier case reports have suggested a link between the increase in anti-SARS-CoV-2 spike IgA and the reactivation of pre-existing IgA vasculitis observed after vaccination (DOI: 10.1016/S2665-9913(21)00211-3). The study suggests caution and suggests physicians to monitor for this possible complication.

Overall, I have one suggestion, since there are quite a few cases reports in this area reporting the new-onset or reactivation of IgA vasculitis, authors could look to present a table listing the key characteristics of those papers and their main conclusion. Adding this table and additional information from similar case studies will greatly enhance the quality and overall utility of this manuscript. The author can do a quick rapid review for finding the case report of IgA vasculitis after SARS-CoV-2 vaccination.

Reviewer 2 Report

Dear authors, 

Overall work here, I believe the authors have done an exemplary job in preparing this manuscript and done a great job for COVID-19. The level of scientific rigor is apparent, and the attention to detail in regard to every aspect of the replication is appreciated. I have a few minor suggestions that the authors might consider, but all of them would be moving forward. 

How did the authors recognize the IgA vasculitis affecting vitals organs? Did the patients get any previous disease history on them before vaccination? 

Round 2

Reviewer 1 Report

Thanks for including the suggestions.